# Thermodynamical versus Logical Irreversibility: A Concrete Objection to Landauer’s Principle

**DOI:** 10.3390/e25081155

**Published:** 2023-08-01

**Authors:** Didier Lairez

**Affiliations:** Laboratoire des Solides Irradiés, École Polytechnique, CEA, CNRS, IPP, 91128 Palaiseau, France; didier.lairez@polytechnique.edu

**Keywords:** thermodynamics, information theory, Landauer’s eraser

## Abstract

Landauer’s principle states that the logical irreversibility of an operation, such as erasing one bit, whatever its physical implementation, necessarily implies its thermodynamical irreversibility. In this paper, a very simple counterexample of physical implementation (that uses a two-to-one relation between logic and thermodynamic states) is given that allows one bit to be erased in a thermodynamical quasistatic manner (i.e., one that may tend to be reversible if slowed down enough).

## 1. Introduction

Entropy was originally defined by Clausius [1] as the state quantity that accounts for heat exchanges and their irreversible features. The state of a system is defined by a set of parameters, the state quantities, such as internal energy, temperature, volume, pressure, quantity of matter, etc., which make it possible to describe the system, i.e., to construct a representation of the system as it appears to our senses (there is nothing else we can access). After Shannon [2], entropy was revealed as the state quantity that quantifies the complexity of this representation considered as a random variable, that is to say, the quantity of information required in unit of bits to encode this representation in the memory of a computer.

The link between information (complexity) and thermodynamics (energy) is neither a metaphor (a figure of speech) nor an analogy (a comparison based on resemblance) nor an interpretation (a personal way to explain something). In all these cases, it would be questionable. However, here, it is absolutely valid and comes from mathematics. Gibbs’ entropy (that of statistical mechanics) is a special case of Shannon’s entropy (the other name for the quantity of information) and the former is directly derived from Clausius’ entropy (for a derivation, see [3]). Hence, the connection. The mathematical relations between the “three entropies” leaves no space for interpretation.

The connection between energy and information makes it possible to understand the functioning of strange machines such as that of Maxwell’s demon [4] and its variations. By acquiring and processing information about the velocities or positions of gas particles, the demon (in the 21st century let us say a computer) is able to establish either a temperature or a pressure difference which, in turn, could produce work. Without a connection between information and energy, the demon’s machinery would either violate the second law of thermodynamics or (more likely) require an energy compensation of unknown origin.

The link between information and energy that comes from Shannon’s information theory is purely mathematical. It has its advantages (rigor) and its disadvantages (abstraction). To overcome the latter, Landauer [5,6], followed by Bennett [7,8], tried to establish this link by using an entirely different method to that of Shannon. Their idea is basically the following. Information, say a set of bits, has necessarily a physical support. So that to be stored and processed, the logical values 0 or 1 of one given bit should “necessarily” (the quotation marks emphasize that it is this precise point which is questioned in this paper) be mapped by a one-to-one relation to the states that can be adopted by a thermodynamical system with a two-minimum (bistable) potential. This one-to-one mapping, known as Launder’s principle, automatically associates information processing with thermodynamics laws. A first corollary is that logical (information) and Clausius entropies behave in the same way, a second is that logical irreversibility (such as erasing one bit of information) implies thermodynamical irreversibility.

Landauer’s principle is actually a conjecture that has been demonstrated in the particular case of a one-to-one physical implementation of a bit. Common objections to this conjecture are mainly conceptual [8]. For example, what exactly “erasing a bit” means has been discussed and the Landauer–Bennett conception has been questioned. The starting point that one bit requires a physical medium to exist has also been questioned [9]. However, this objection is more philosophical than scientific. It deals with the meaning of “existence”. Personally, I am a materialist who thinks that even abstractions need physical support, at least in the form of our brain thinking them. However, this question ultimately boils down to discussing whether or not something exists outside of our perception. As interesting as it is, this question is not the responsibility of science, whose theories are founded and validated by experiments [10].

The purpose of this paper is to propose a concrete (as opposed to conceptual) objection that comes as close as possible to the requirements of what an ERASE operation should be according to Landauer. Establishing Landauer’s conjecture as a principle presupposes its generality that can be accepted until proven otherwise. In other words, whereas it is not possible to definitely prove that it is true, it is possible to prove it is false by finding a counterexample. This is the concern of this paper that presents a bistable bit linked (by a two-to-one surjective relation) to a monostable thermodynamical potential. This two-to-one implementation allows logical irreversibility to occur in a thermodynamical quasistatic manner, which may, therefore, tend to be reversible if slowed down sufficiently.

## 2. Irreversibility

### 2.1. Logical Irreversibility

Logical operations take one or more bit-values as input and produce a single bit-value as output. They are logically irreversible if the probability of a given output value differs from that of the input. Reversible logical operations preserve the quantity of information, whereas irreversible operations do not. In the case where the initial value of one given bit is known, the operation RESET TO 0 [5] (equivalent to ERASE [7]) is logically irreversible because two possible initial values (0 or 1) lead to a single result (0). Once the operation has been performed, the information on the initial value of the bit is lost.

Note that erasing a set of bits decreases (or leaves constant) their statistical entropy, e.g., a set of bits with random equiprobable values and maximum entropy has zero entropy once all values are reset to 0. So that associating the corresponding logical irreversibility to thermodynamical irreversibility is not straightforward as the latter is most easily associated with an increase in the entropy of the system (one can think for instance of spontaneous processes that occur at constant internal energy).

### 2.2. Thermodynamical Irreversibility

Consider a system that can be found in two different states, say A and B. In thermodynamics, a process which would consist in bringing the system from A to B (A→B) is said to be reversible (or irreversible) if there is a restore process (B→A) that allows the system to be returned to its initial state, and such that at the end of the whole cycle (A→B→A) the net quantity of heat *Q* produced by the system and dissipated into the surroundings is zero (or strictly negative, sign with respect to the system). In most cases, heat is not the quantity we are interested in. Rather, it would be its complement
(1)$$E_{\mathrm{cost}}=-Q,$$
that is, the sum of all other forms of energy supplied to the system.

First, note that whether a process is reversible or irreversible is not an inherent property of states A and B, but a property of the path that is taken. In reality, no thermodynamical process is exactly reversible, perpetual motion does not exist, and a cycle is always accompanied with the dissipation of energy in the form of a net quantity of heat passing from the system to its surroundings. This is formalized by the second law of thermodynamics which has never been faulted and will not be by this paper either.

To illustrate this point (see Figure 1), let us consider a unit amount of gas in contact with a temperature reservoir at temperature *T*. A typical example of a reversible cycle is that of the isothermal expansion/compression from volume *V* to 2V by the means of a piston. If the temperature of the gas is effectively kept constant throughout the process, so is its internal energy. Due to the conservation principle of energy, at every moment the gas provides mechanical work 
dW but draws exactly the same quantity of heat dQ from the environment. At the end of the expansion stage: Q=−W=Tln(2V)−Tln(V)=Tln2 (where *T* is expressed in joules). The cycle is closed by a restore process that involves exactly the same amount of heat and work but with opposite signs. So that $E_{\mathrm{cost}}=0$. This is an ideal reversible situation that is never reached.

There are two fundamentally different categories of irreversible processes (therefore, two categories of process altogether). Consider the above expansion with a piston. From a general rule in physics there exists an unavoidable delay between cause (expansion) and effect (thermalization), which here implies the inequality Tln2≥Q.

The second law of thermodynamics is twofold: (1) it identifies ln2 (in this example) as being the difference ΔS of a state quantity, namely, the entropy *S* of the system that only depends on A and B, but not on the process; (2) it tells us that TΔS≥Q (in all cases), so that:(2)TΔS=Tln2≥Q,
where the equality holds for an infinitely slow process allowing an infinitely small delay. The second law of thermodynamics says absolutely nothing more than that.

Note that the equality in Equation (Equation 2) gives us the only way to measure ΔS, in particular by using an infinitely slow process (i.e., reversible) to go back to the initial state (B→A). In our case, this corresponds to the compression of a piston from B to A that requires mechanical work and provides heat to the surroundings (ideally both with the same absolute value as for the expansion but with opposite sign). Using this restore process, the energy cost at the end of the cycle (A→B→A) is:(3)$$E_{\mathrm{cost}}\geq 0$$
which means that there is no conceptual impossibility for this energy cost to be as small as desired by slowing down the process, which is then said to be quasistatic. Processes belonging to this first category can be considered as potentially reversible. This includes all those which experience only friction and always remain under control.

Another class of thermodynamic processes are those that are inherently irreversible because they are out of control. An example is the adiabatic free expansion (see Figure 1) of a gas without a piston: no heat and no work are exchanged with the surroundings (it is adiabatic and there is no piston to capture the work). However, something happens because work has to be done (this time with a piston) to restore the system to its initial state. This restoring process is the same as in the previous category, so that the energy balance of such a cycle is:(4)$$E_{\mathrm{cost}}\geq T\ln2,$$
which means that Tln2 (the entropy difference between A and B expressed in temperature units) is a lower limit for $E_{\mathrm{cost}}$ that can never be bypassed.

A point which deserves to be underlined in order to avoid a misinterpretation of what follows, is that the assertion that a process belongs to one category or another is not a violation of the second law of thermodynamics. The two categories are both consistent with it.

Landauer’s principle claims that all physical implementations of the operation RESET TO 0 (or ERASE) correspond to processes that belong to the second category (inherently irreversible like the adiabatic free expansion). The aim of this paper is to provide a counterexample that belongs to the first (potentially reversible if slowed down enough, like the monothermal expansion).

## 3. Erasers

### 3.1. Landauer’s Eraser (One-to-One Implementation)

For some reasons (that are not challenged in this paper), Landauer [5] considers that the two logical states of a bit, 0 and 1, cannot be physically implemented with the two states A (volume *V*) and B (volume 2V) of the previous section. A correct physical implementation must fulfill two conditions (see [5] p. 184):(1)states 0 and 1 must be stable;(2)the operation RESET TO 0 must correspond to the same physical process whatever the initial state.

Next, Landauer claims (here is the very point challenged in this paper) that the only possible way to fulfill these two conditions is to realize a one-to-one mapping between the two bit-values and two stable thermodynamical states separated by an energy barrier, such as a particle in a bistable potential.

Note that as is, the bistable potential seems to be not convenient to fulfill the second condition, because there is nothing to do to RESET TO 0 in case the initial state is already at 0, while an energy barrier must be crossed otherwise. To overcome this problem, Landauer imagines the following functional procedure that follows three stages (Figure 2):(1)lower the energy barrier down to a value smaller than the thermal energy *T*, leaving the system to a “standard” (S) state (Consider a single particle in a diathermal box in contact with a temperature reservoir. Even if this particle is alone, its temperature is well defined by the multiple collisions with the wall of the box. Let us assign bit value 0 when the particle is in the left side and bit value 1 when the particle is in the right side. The logical states are stable and well defined only when a barrier exists (higher than thermal energy) between the two sides. The S-state corresponds to the situation where the barrier is removed);(2)apply a small energy bias in the desired direction in order to drive the particle into the desired state;(3)put up the barrier and remove the bias.

The point is that during the first stage, the probability density of the particle leaks from its initial potential well to fill both [7]. This leakage occurs in an out-of-control and irreversible thermodynamical manner, because putting up the barrier at the end of this stage would not necessarily return the particle back to its initial well. Like free expansion, stage 1 occurs without energy exchange with the surroundings, contrary to the rest of the procedure that can be quasistatic, amounts to an isothermal compression, and dissipates at least Tln2 (as heat) to the surroundings. In the rest of the argument, Bennett [7] proves that the initial logical state can be restored to the initial value (0 or 1) by a WRITE operation that can be performed in a quasistatic manner. So that the whole cycle (ERASE then RESTORE) costs at least Tln2 (as in Equation (Equation 4)). It follows that this physical implementation of the irreversible logical operation RESET TO 0 is a process of the second category, that is to say, thermodynamically intrinsically irreversible (Equation (Equation 4)).

The direct physical implementations of a bit by using a bistable potential has been experimentally achieved. Berut et al. [11] trapped a colloidal particle with a double-beam optical tweezer. Hong et al. [12] directly worked on magnetic memory at the nanoscale. This type of experience is unquestionably very difficult and the state of the art. The authors actually measure a lower energy bound equal to Tln2 to move the bit from one state to the other. So that the irreversible and out-of-control “leakage” invoked by Bennet to erase one bit seems unavoidable. However, Landauer’s principle is stronger than that. It states that it is unavoidable whatever the physical implementation.

### 3.2. Counterexample (Two-to-One Implementation)

The counterexample I propose is based on the fact that the irreversible leakage from one potential well to the other of Landauer’s eraser cannot occur if there is only one potential well: that is to say, two logical states corresponding to one single thermodynamical state. It remains to find a physical implementation allowing this.

Let us fill a diathermal gas container below a piston at atmospheric pressure while the piston is at the position of maximum expansion. Then, close the container. This thermodynamical system is monostable when the piston is up (Figure 3). Let us link the piston to a connecting rod, a crankshaft, and a pulley of radius 1. A frequency divider is obtained with a belt and another pulley of radius 2 equipped with a crank, so that to the single stable position of the piston there correspond two stable positions of the crank (up and down in Figure 3 if the belt is initially closed while the two pulley angles are zero). The two crank positions define a bit whose thermodynamics depends on: (1) the expansion/compression of the gas; (2) the friction of the transmission. As both can be quasistatic, operations on this bit are too.

Before investigating bit operations, note that (1) due to conservation of energy, the height of energy barriers for the crank (logical barrier) increases linearly with the gear ratio crank/crankshaft, whereas that of the piston is constant (thermodynamical barrier); (2) the gear ratio can vary continuously by using a so-called “continuously variable transmission” mechanism, say, for instance, a conical pulley for the crank. So that there is no conceptual impossibility for this variation to be performed as slowly as desired in a fully controlled and quasistatic manner. It follows that, while the bit is initially at an equilibrium position (either 0 or 1), the gear ratio can be decreased enough so that the logical barrier becomes smaller than the thermal energy *T* (see Figure 4). Then, due to fluctuations of pressure below the piston, the position of the crank can fluctuate in any position between 0 and 2π, leading to an undetermined bit value. We, thus, obtain a soft potential well (standard bit-state S), as in the papers of Landauer and Bennett [5,7].

The RESET TO 0 operation can be performed by the following sequence which is copied from that of Landauer-Bennett:(1)put the gear ratio to a small enough value so that the bit is in the S-state;(2)set the crank to the desired position 0 (by applying in a quasistatic manner a force similar to the bias in the Landauer–Bennett implementation);(3)put the gear ratio back to 2.

This sequence is analogous to that of Landauer–Bennett (so that the bit is erased), but there is a major difference due to the two-to-one implementation. In the S-state, the crank can move without modifying the position of the piston. The bit position (logic) and piston position (thermodynamic) are practically uncoupled. Another way to say the same thing is that at the end of the first stage putting up the energy barrier does not necessarily return the system to the same logical state (logical irreversibility), but necessarily leaves the system to the same thermodynamical state (thermodynamical reversibility) because there is only one potential well.

As the other stages of the sequence do not involve rotation, the overall operation is performed without any change in the thermodynamical state, nor energy dissipation (except that of the friction of the transmission, that can be as small as desired). Note that Shenker [13] (Figure 5 in his paper) proposed another mechanism allowing the coupling/uncoupling logic and thermodynamic parts. However, the discontinuous procedure for the operation does not permit it to be quasistatic, as explained by Bennett [8].

The bit implementation proposed here avoids this issue. So, although it is logically irreversible, the procedure with this implementation may be thermodynamically quasistatic. This procedure fulfills the two conditions stated by Landauer (1—two stable bit-states, 2—the same procedure whatever the initial bit-state) and obeys the same sequence as that of Landauer (1—lower the barrier; 2—apply a bias; 3—raise the barrier). So that if these criteria are correct, this two-to-one implementation is also correct from a computational point of view. This implementation permits the bit to really be erased, at least as much as that of Landauer, but this time in a quasistatic manner, allowing $E_{\mathrm{cost}}$ to be down to the Tln2 limit, provided the operation is performed slowly enough. Here, “slowly enough” is in comparison with the rate of thermalization of the gas below the piston (including the heat transfer through the container wall) and fundamentally means “allowing $E_{\mathrm{cost}}$ to be smaller than Tln2”. The characteristic fluctuations rate (or the relaxation rate) of the system (here the gas) could be viewed as a practical lower-boundary limitation for the rate of the process. However. actually it is not, for two reasons: (1) the height the energy barrier can be (in principle) is as high as desired (it only depends on the ratio of the pressures between the two extreme positions of the piston); (2) in the S-state the bias value can be increased as much as desired (in principle) as the process is slowed down.

Note that the above physical implementation, here exposed for binary logic, could be easily extended to multivalued logic [14] by increasing the maximum value of the gear ratio *r*. For instance, r=3 would allow three logical states, etc.

## 4. Maxwell’s Demon, Szilard’s Engine, and Ratchets

Maxwell was far ahead of his time and was the first to understand the link between energy and information. He imagined [4] a gas in an insulating container separated in two parts along the *x*-axis by a thermally insulating wall having a small door. A demon is able to measure the velocity component $v_{x}$ of molecules and open the door, allowing faster molecules to go from *A* to *B* while slower ones can only go from *B* to *A*. This results in a decrease in the entropy of the system or equivalently in a temperature difference between the two compartments, which can eventually be used for running a thermodynamic cycle and producing work.

A simplified version of this device is due to Szilard [15], the system is made of a single “gas” particle submitted to thermal motion in a box divided into two parts (say left and right). The demon puts a wall on the middle, then measures where the particle is, places a piston on the opposite side, and removes the wall so that the pressure can produce work on the piston. Today, Szilard’s engine is no longer a curiosity. It is at the basis of some experimental realizations of Feynman’s ratchet [16] at a molecular level [17,18,19] with potential interesting applications.

How can these machines work in accordance with the second law of thermodynamics? Landauer’s principle is often presented as the key point for their understanding. Let us examine this.

### 4.1. Energy, Entropy, and Information

Energy is a strange physical quantity. It is a universally used concept, but there is no definition of what exactly energy is. Actually, energy is an abstraction only defined by a conservation principle. This is explained by Feynman in his physics lessons [20]. In thermodynamics, this conservation principle originates from the experiments of Joule [21], who produced heat (in calorie) by providing mechanical work (in Nm) and observed that both quantities are proportional, so that by using the same unit (joule) one can introduce a quantity (namely the internal energy) that is constant for an isolated system. Then, each time this principle of conservation seems to be violated, it suffices, to conserve it, to declare that a new form of energy has been discovered.

The introduction of the concept of entropy with the second law of thermodynamics is quite similar. This law can be stated as: (1) there exists a form of internal energy proportional to the temperature *T* and to a state-quantity *S* named entropy such that TΔS is the heat exchanged for a reversible process; (2) the entropy of the state of a system cannot decrease at no cost in energy. Each time this law seems to be violated, it suffices, to preserve it, to declare that we have missed something in performing the energy balance.

Maxwell is also the person who wrote, well before Shannon’s theory of information, “The idea of dissipation of energy depends on the extent of our knowledge” [22]. So, it is clear that in his mind the missing term in the energy balance lies in the knowledge (or information) necessary for the demon to operate. Information is a new form of energy.

After Shannon, things become clearer. The link between entropy and probabilities was made by Boltzmann, Planck, and Gibbs [23,24,25], leading to the equality: $S=\sum_{i}p_{i}\ln\left(1/p_{i}\right)$, where $p_{i}$ is the probability for the system to adopt the microstate *i*. Then, Shannon [2] demonstrated that this quantity is also the average number of bits required to encode and store a representation of the microstate of the system (i.e., the minimum requirement to treat this information). The link between information and energy is made and Maxwell’s demon machine has no more mystery. Except we do not know exactly where inside the demon the energy dissipation is happening. If we want to be more precise, as a demon does not exist, we must first specify what we are going to replace it with. We must enter into the details of the physical implementation of the ratchet. However, before this, let us first note that there is absolutely no reason for the “location” of the dissipation to be universal. So that with regard to what a theory should be, i.e., not an explanation but rather an economy of thought [10,26], it is not certain that we gain much by doing this.

### 4.2. Landauer–Bennett vs. “Shannon Only” Interpretations

Following Bennett [7], Szilard’s demon is replaced by a Turing machine and the measure is a COPY of the particle position (left or right) to one bit (0 or 1) of the memory buffer of the machine to the state on which the rest of the process depends. To run cyclically, the COPY operation is actually an OVERWRITE, that can be split into ERASE (i.e., RESET TO 0) then WRITE. Hence, the answer: the expansion of the gas produces mechanical work equal (at best) to Tln2, but the ERASE costs (at least) the same quantity (Equation (Equation 4)). In Bennett’s mind, the place where dissipation occurs is then well identified (ERASE operation). Landauer’s principle claims the generality of this. Two objections can be made to this reasoning.

The first objection is that splitting OWERWRITE is not necessary. The overwriting can be performed directly according to the same mechanism as in Figure 2 (i.e., independent of the initial state) but with a final state which depends on the measurement and which is not always equal to 0. For a cyclic process, overwriting the previous measurement by the last one does not change the entropy of the system because both measurements have the same probability distribution. In this case, OVERWRITE is logically and thermodynamically (statistically) reversible, even in the framework of Landauer’s physical implementation. Introducing an irreversible ERASE operation is artificial.

Note that this objection is different to that of Earman and Norton [27] who attempt to replace ERASE by reversible operations. The authors introduce a conditional operation (IF) that is equally logically irreversible, as explained by Bennett ([8] p. 505) in his refutation.

The second objection directly comes from the counterexample given in this paper: the reasoning falls down if ERASE can be achieved in a quasistatic manner as is shown here.

Understanding of Szilard’s engine in the framework of Shannon’s information theory is different. Because the phase space of the gas is discrete and finite, a given microstate corresponds to a given value of an integer random variable with a finite support. The gas is a random source of information that can be encoded by using a number of bits per word (per microstate) equal to the Shannon’s entropy (namely, the quantity of information or the uncertainty about the source). As Shannon’s entropy of microstates distribution is the same as Gibbs’ entropy, that is the same as Clausius’ entropy, it follows that reducing the uncertainty in the source by a factor of 2 (making the economy of one bit) has necessarily an energy cost at least equal to Tln2 (according to the second law used here, exactly as it was also in Landauer’s eraser approach). This is exactly what is performed when Szilard’s demon puts up the wall prior to any measurement. However, this should be seen as part of a cycle with an arbitrary beginning and end. So that the cost has not to be paid immediately but either by the rest of the process necessary to close the loop or its equivalent belonging to the previous cycle. This solution *à la* Shannon does not enter in the detailed mechanism of the demon’s black-box, thus leaving the space for specific implementations. It is free of any physical support for information because the entropy of the source (the emitter) does not depend on the presence or absence of a receiver (the physical support). For instance, Brillouin [28] outlines that the demon needs light to see where the particle is (measurement) and that the energy needed for this light emission will prevent violation of the second law.

What we know from Shannon, is that Tln2 must we paid somewhere for Szilard’s engine to work, but that this cost can be everywhere in the demon and is not necessarily due to an ERASE operation: (1) because ERASE is not unavoidable; (2) because ERASE can be performed in a thermodynamical quasistatic manner.

## 5. Concluding Remark on Computing Power Limits

Computing requires ERASE operations. As a consequence, Launder’s eraser energy cost (Tln2) is often considered as an absolute quantity that limits the computing power. The bit-implementation given in this paper shows that this idea is not correct: logical irreversibility does not necessarily imply thermodynamical irreversibility.

The question is not whether a computer can be built by using such a mechanical implementation (clearly it should not), but rather whether other two-to-one implementations would allow the same result. This cannot be excluded, in particular in cases where the information is not processed by computers but by biological systems. Launder’s principle that involves a one-to-one implementation is likely very (may be the most) common, but it is not general (the counterexample demonstrates this). Szilard’s engines need at least Tln2 per cycle to work in agreement with the second law whatever way the engines are physically implemented. However, computing is not only dedicated to these engines. Following Landauer [6], there are no unavoidable energy consumption requirements per step in a computer provided reversible computation is performed. This article shows that this assertion can be extended to irreversible computation.

## Figures and Tables

**Figure 1 entropy-25-01155-f001:**
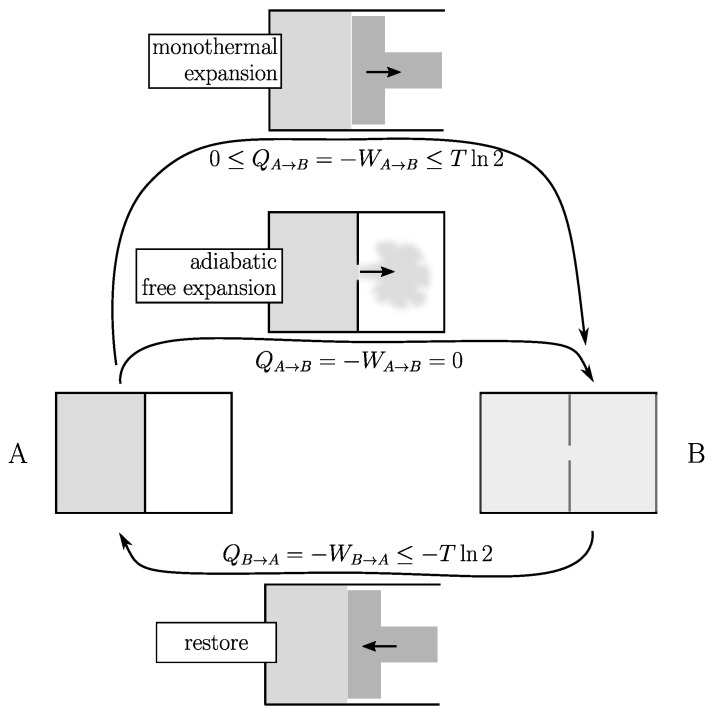
Expansion of a unit amount of gas from volume *V* (state A) to 2V (state B) at the same temperature *T* (in joules). (1) Monothermal expansion with a piston (top): the gas produces work *W* and pumps heat *Q*. (2) Adiabatic free expansion (middle): no heat and no work are exchanged with the surroundings. In both cases the cycle is closed using the same restoring process (bottom). In the first case, the net energy cost $E_{\mathrm{cost}}=-Q$ of a cycle can be as small as desired (quasistatic, Equation (Equation 3)). In the second case, it has a lower limit of Tln2 (Equation (Equation 4)).

**Figure 2 entropy-25-01155-f002:**
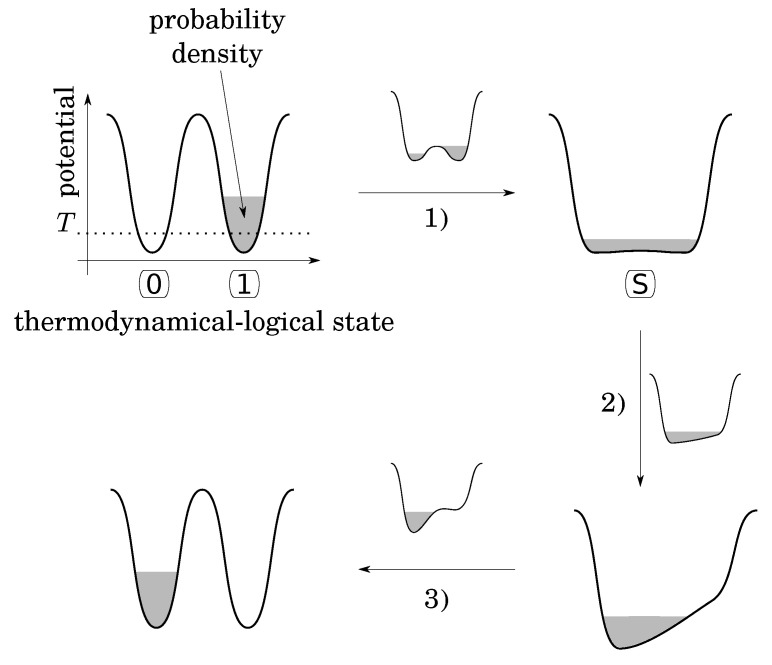
Landauer’s functional procedure to physically implement the RESET TO 0 (ERASE) logical operation by the means of a thermodynamical bistable potential with a tunable barrier and a bias. Here, the bit is initially set to 1 but the same procedure would apply if the was were set to 0.

**Figure 3 entropy-25-01155-f003:**
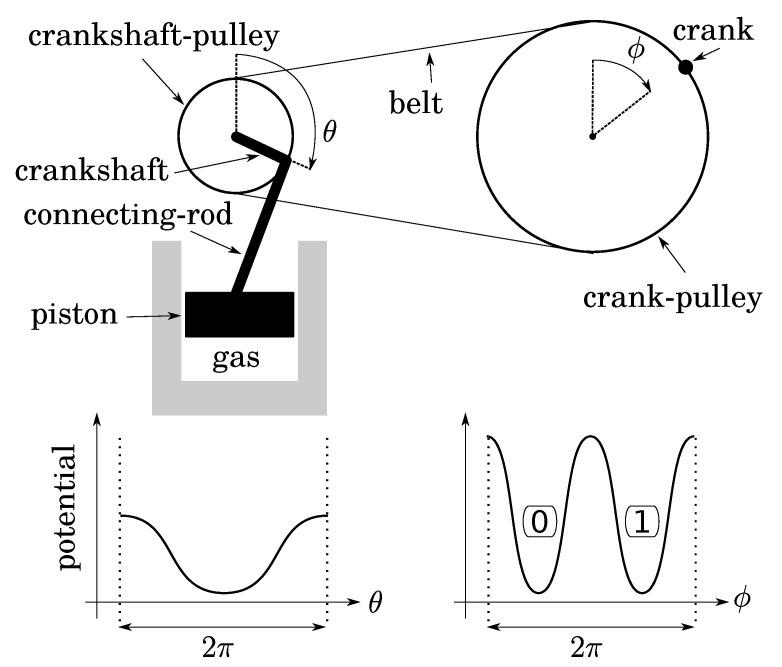
Two-to-one implementation of a bit: quasistatic isothermal compression/expansion of a gas is performed with a transmission of gear ratio 2 (crank–pulley/crankshaft–pulley). The two stable positions of the crank (to which are assigned bit values 0 and 1) correspond to a single stable position of the piston.

**Figure 4 entropy-25-01155-f004:**
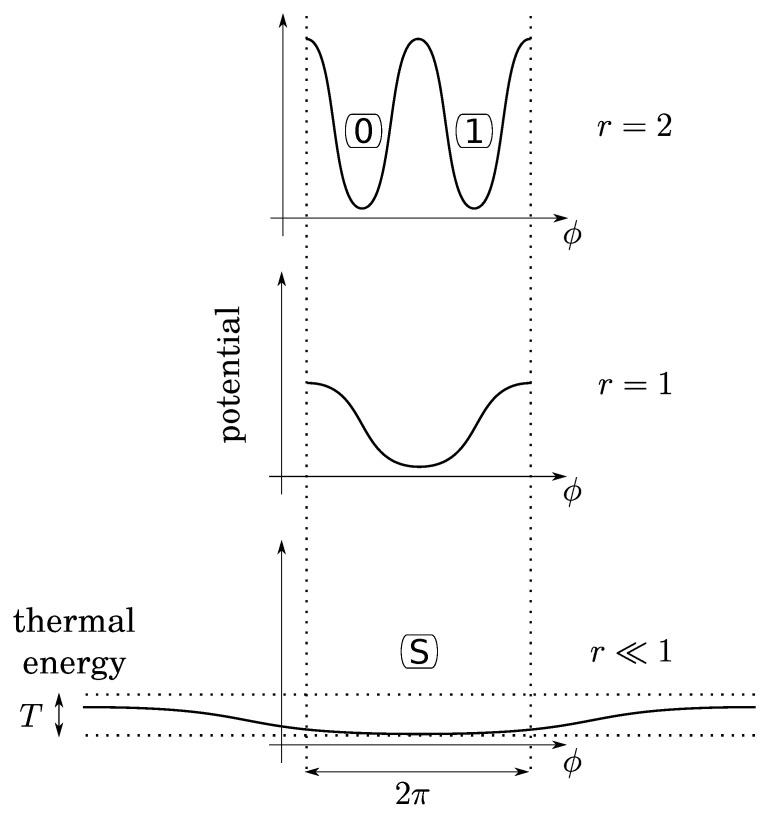
The gear ratio *r* can be small enough so that thermal fluctuations of the gas below the piston permit the crank-angle (ϕ) to be in any position. This soft potential well determines a third state (S for “standard”) for a virgin bit.

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
