# Peer review of "Thermodynamical versus Logical Irreversibility: A Concrete Objection to Landauer’s Principle"

_entropy, 2023, doi:10.3390/e25081155_

Round 1
Reviewer 1 Report
Thermodynamical versus logical irreversibility: a concrete objection to Landauer’s principle
By Didier Lairez
Dear Editor,
I had the pleasure to review this manuscript.
The author of this article tackles an important topic involving the relationship between the information and thermodynamics.
The author makes a nice introduction and acknowledges the previous studies including the experimental demonstration of the Landauer‘s principle.
In this article though, the author makes the case that the generality of the Landauer’s principle could be questioned if one example of its violation could be shown.
The author then sets out to attempt to prove exactly this in the present article.
The approach taken by the author involves showing that logical irreversibility may tend to be reversible if the process is slowed down sufficiently. Others have discussed this approach and argument and it is not new.
The slowing down sufficiently is a vague term that brings in discussion an observer and it departs from the stated conditions that the bits must be stable and the logical operations must take place faster than the relaxation time of the system.
For example, if a bit of memory undergoes thermal fluctuations with a relaxation time of 1 ms, then any observer would see no memory state if the observation time is larger than 1 ms. On the other hand, if the observer takes a reading in a microsecond, then he/she would be able to extract the 0 or 1 state, depending on the state at time of observation.
Unfortunately this is not an example of violation of Landauer’s principle. Using extreme cases such as unstable bit states, infinite time, etc is not a valid argument because it requires new physics such as non-equilibrium statistics, etc.
The second issue I have with this article is the fact that the erase operation requires a reset to 0 or 1 state, as defined by Landauer. However, a more recent article clearly states that a true erase is to completely remove the state from existence, i.e. there is no 0 or 1 state left and the entropy is truly reduced (https://pubs.aip.org/aip/adv/article/9/9/095206/1076232).
The claim that information states need material support is true from a technical point of view, but when talking about information as abstract mathematical states, these can exists without any material support and they have entropy associated to them. The link between their physical entropy and the Shannon information entropy was given here: https://pubs.aip.org/aip/adv/article/9/9/095206/1076232
This can manifest without material states, and one day it may even be possible to create devices based on this concept, i.e. storing data in non-material states.
To summarize, I find this article rather instructive, pedagogical and useful for a number of readers. I do not agree with its conclusions, but in this field of science the best is to open up the debate and allow all ideas and arguments to be debated. For this reason, I recommend publication of this article, after some optional comments addressing the comments made above.
Author Response
*************
First issue: about "slowing down sufficiently..."
In the context of the paper, "sufficiently slow" means "allowing Ecost to be smaller than Tln2".
However, the characteristic fluctuations rate (relaxation rate) of the system could be viewed as a practical lower-boundary limitation for the rate of the process. But actually it is not for two reasons:
1) the height the energy barrier can be (in principle) a high as desired (it only depends on the ratio of the pressures between the two positions of the piston);
2) in the S-state the bias value can be increased as much as desired (in principle) as the process is slowed down.
This discussion has been added (see lines 234-243) in the revised version.
*************
Second issue: it is actually twofold.
1) ERASE is not RESET TO 0
This point actually concerns another objection that could be done to the Landauer's principle. Criticisms of Launder's principle based on this (ERASE is not RESET TO 0) are conceptual.
But the aim of this paper is to propose a concrete objection (title) that matches as close as possible to the requierements of what should be an ERASE operation according to Landauer.
2) Does a bit need a physical medium to exist?
This is another conceptual objection that could be done to the Landauer's principle. But this objection is more philosophical than scientific.
It deals with the meanning of "existence".
Personnally, I am a materialist who think that even abstract mathematics needs physical support at least in the form of our brain thinking them. This question ultimately boils down to discussing whether or not something exists outside of our perception.
As interesting as it is, this question is not the responsibility of science whose theories are founded and validated by experiments.
In order to clarify the context of the paper, in the introduction these points of the discussion are now mentioned in a new paragraph (see lines 44-53) in the revised version, with the very interesting reference that you brought to my attention.
Reviewer 2 Report
The paper needs a very serious revision.
Remarks:
1."Entropy was originally defined by Clausius [ 1 ] as the state quantity that accounts for heat exchanges and their irreversible features".
The statement is obscure, I think that the author meant "Entropy was originally defined by Clausius [ 1 ] as the state function that accounts for heat exchanges and their irreversible features".
2. In the text:
"The state of a system is defined by a set of parameters, the state quantities, such as internal energy, temperature, volume, pressure, quantity of matter etc, which make it possible to describe the system, i.e. to construct a representation of the system as it appears to our senses (there is nothing else we can access)".
In this statement thermodynamic state variables (temperature, pressure, volume, etc.) and thermodynamic state functions (Internal energy) are mixed. The thermodynamic state variables and thermodynamic state functions should be clearly distinguished.
3. Caption to Figure 3:
Figure 3. Two-to-one implementation of a bit : quasistatic isothermal compression/expansion of a gas is performed with a transmission of gear ratio 2 (crank-pulley/crankshaft-pulley), so that to the single stable position of the piston corresponds two stable positions of the crank to which are
assigned bit values 0 and 1".
It should be:
Figure 3. Two-to-one implementation of a bit : quasistatic isothermal compression/expansion of a gas is performed with a transmission of gear ratio 2 (crank-pulley/crankshaft-pulley), so that to the single stable position of the piston corresponds to the two stable positions of the crank to which are assigned bit values 0 and 1"
4. In the text: "So that there is no conceptual impossibility for this variation to be done as slowly as desired in a fully controlled and quasistatic manner. It follows that, while the bit is initially at an equilibrium position (either
0 or 1), the gear ratio can be decreased enough so that the logical barrier becomes smaller than the thermal energy T (see Fig.4). Then, due to fluctuations of pressure below the piston, the position of the crank can fluctuate in any position between 0 and 2π, leading to an undetermined bit value. ".
This is the main point that I did not understand: if thermal fluctuations are so large that the logical state is uncertain, it seems the notion of temperature is not defined and could not be introduced. Local thermodynamic equilibrium is necessary for introducing the notion of the thermodynamic temperature.
The English should be deeply edited.
Author Response
*************
Points 1 and 2:
I do not agree with these remarks.
There is fundamentally no difference between VARIABLES and FUNCTIONS both are physical QUANTITIES. But, all these quantities are not independent each other. Once a set of independent quantities (with a maximum cardinality) is chosen as coordinates to represent the state of the system, these quantities automatically become VARIABLES and the others FUNCTIONS. But another representation would lead to another set of VARIABLES and another set of FUNCTIONS. We can pass from a given representation to another by Legendre's transforms, allowing the VARIABLES to become FUNCTIONS and vice versa.
Therefore, entropy (S) is not intrinsically a state function and temperature (T) is not intrinsically a variable.
*************
Point 3:
Thank you for this careful reading.
Actually, the figure caption was not clear and has been modified in the revised version.
*************
Point 4:
This point concerns the "S-state" of Landauer-Bennett implementation which is also included in the physical implementation that I propose.
In the framework of Landauer-Bennett implementation:
Consider a single particle in a diathermal box in contact with a temperature reservoir. Even if this particle is alone, its temperature is well defined by the multiple collisions with the wall of the box. Let us assign bit value 0 when the particle is in the left side and bit value 1 when the particle is in the right side. The logical states are stable and well defined only when a barrier exists between the two sides. The S-state corresponds to the situation where the barrier is removed.
In the framework of a two-to-one implementation:
The gas below the piston is also in a diathermal box in contact with a temperature reservoir. So that the temperature is always well-defined.
To clarify this point a footnote has been added (line 159) and "diathermal" added line 188 is the revised version.
Round 2
Reviewer 1 Report
I am satisfied with this version of the manuscript.
Author Response
Thank you for your report.
Note that my answer about the rate of the process has been slightly improved in the new version (see lines 237-240).
Reviewer 2 Report
Remarks:
1. Author Reply
"There is fundamentally no difference between VARIABLES and FUNCTIONS both are physical QUANTITIES. But, all these quantities are not independent each other. Once a set of independent quantities (with a maximum cardinality) is chosen as coordinates to represent the state of the system, these quantities automatically become VARIABLES and the others FUNCTIONS. But another representation would lead to another set of VARIABLES and another set of FUNCTIONS. We can pass from a given representation to another by Legendre's transforms, allowing the VARIABLES to become FUNCTIONS and vice versa. Therefore, entropy (S) is not intrinsically a state function and temperature (T) is not intrinsically a variable".
This is a deeply erroneous point of view. From the pure mathematical point of view there is no difference between thermodynamic variables and functions. However, there exists also the logic of physics, and it implies the distinction between thermodynamic variables and functions. Let us explain: physical approach insists on the difference between intrinsic (mass independent) and extrinsic (mass dependent) physical properties. All of the thermodynamic functions, namely the internal, Gibbs and Helmholtz energies and also entalpy are extrinsic physical values; whereas, temperature and pressure are intrinsic physical values. And this difference is essential. Mathematics neglects this distinction, but it is of a primary importance for physics.
2. I will ask the author to extent the reported approach for the computing devices based on many-valued logic, see:
Generalization of the Landauer Principle for Computing Devices Based on Many-Valued Logic, Entropy 2019, 21(12), 1150
The English is OK.
Author Response
***************
Point 1:
This debate is likely out of the scope of this paper.
However, I would like to answer to the sentence "deeply erroneous point of view" (sic).
You would probably agree that functions and variables are mathematical objects. By using them, physicists are obliged to adopt the meaning that was given to them by mathematicians. This is not a question of point of view.
Mathematics is the tool and language common to all fields of science. It would be a very bad idea for physicists to adopt a particular meaning for the words of the mathematical vocabulary.
***************
Point 2:
Thank you for the reference. It has been added to the paper with a small paragraph about many-valued logic (end of section 2, lines 245-247 in the new version).
Round 3
Reviewer 2 Report
The paper is publishable.